# COVID-19: A Semantic-Based Pipeline for Recommending Biomedical Entities

**Marcia Barros[1,2], Andre Lamurias[1], Diana Sousa[1], Pedro Ruas[1]** and **Francisco M. Couto[1]**

[1]LASIGE, Faculdade de Ciências, Universidade de Lisboa
[2]CENTRA, Faculdade de Ciências, Universidade de Lisboa

`mcbarros@fc.ul.pt, alamurias@lasige.di.fc.ul.pt,`
`dfsousa@lasige.di.fc.ul.pt, psruas@fc.ul.pt, fcouto@di.fc.ul.pt`

## Abstract

With the increasing number of publications about COVID-19, it is a challenge to extract personalized knowledge suitable for each researcher. This work aims to build a new semantic-based pipeline for recommending biomedical entities to scientific researchers. To this end, we developed a pipeline that creates an implicit feedback matrix based on Named Entity Recognition (NER) on a corpus of documents, using multidisciplinary ontologies for recognizing and linking the entities. Our hypothesis is that by using ontologies from different fields in the NER phase, we can improve the results for state-of-the-art collaborative-filtering recommender systems applied to the dataset created. The tests performed using the COVID-19 Open Research Dataset (CORD-19) dataset show that when using four ontologies, the results for precision@k, for example, reach the 80%, whereas when using only one ontology, the results for precision@k drops to 20%, for the same users. Furthermore, the use of multi-fields entities may help in the discovery of new items, even if the researchers do not have items from that field in their set of preferences.

## 1 Introduction

The research literature is the main form of dissemination for scientific works, growing by the minute. Platforms such as PubMed[1], account for more than 30 million articles related to biomedical literature. The emergence of new topics of investigation with particular interest for modern society, such as COVID-19 (Organization et al., 2020), leads to an even faster increase in the publication rate. The scientific literature contains vast and essential information about biomedical entities engaged in COVID-19 processes. However, it is difficult for the researchers to read all the papers and keep up with all the new topics suitable for their research.

Given the importance of COVID-19 related topics, the Allen Institute for AI, in collaboration with The White House Office of Science and Technology Policy, the National Library of Medicine, the Chan Zuckerburg Initiative, Microsoft Research, and Kaggle, collected and released the first version of COVID-19 Open Research Dataset (CORD-19)[2] (Wang et al., 2020a). The main goal of this dataset is to help in the development of new tools for the extraction of relevant information in the fight of COVID-19 disease.

One of the main techniques applied to extract information from the research literature is Named Entity Recognition (NER), followed by Relation Extraction (RE). NER consists of recognizing entities mentioned in the text by identifying the offset of their first and last character. There is an extensive work done on biomedical NER, regarding all type of entities, such as chemicals (Lamurias et al., 2013) and human phenotypes (Lobo et al.). RE aims to identify a relation between entities mentioned in a given document or text window. Regarding biomedical RE, the research is focused not only on extracting but also on classifying the relationship between biomedical entities, ranging from phenotype-gene relations (Sousa et al., 2019) to chemical-chemical interactions (Herrero-Zazo et al., 2013).

Recommender Systems (RS) are tools which allow recommending items of interest to a user, based on the similarity between her/his preferences and the preferences of other users - Collaborative-filtering (CF), or based on the similarity of the

---

[1]https://pubmed.ncbi.nlm.nih.gov/

[2]https://www.kaggle.com/
allen-institute-for-ai/
CORD-19-research-challenge

items this user already liked - Content-Based (CB). Hybrid RS may be created to solve intrinsic challenges of the previous RS approaches, such as the cold start problem for new items in CF and new users in both CF and CB. Cold start refers to a new user without any item rated or a new item without any rate by the users. The users' ratings may be explicit, for example, using a star classification system, or implicit, in which case the ratings are inferred from the user's interaction with the items, for example, by buying or seeing them. The implicit rating may be binary, for example, 1 if a user saw a movie, 0 if she/he did not see a movie, or it may be a different measure, for example, the duration that a user watches a video (Ricci et al., 2015).

RS are most frequently used for recommending items such as movies, books, or e-commerce products. RS approaches have been applied most recently to recommend the most appropriate research items for each researcher (Ortega et al., 2018; Barros et al., 2019, 2020). These efforts consist mostly of developing recommendation datasets of implicit feedback for various scientific fields, such as Chemistry and Artificial Intelligence, developed by mining the scientific literature. The goal of these datasets is to recommend scientific items for the research, for example, Chemical compounds, based on their past interests and their peers' interests.

This paper proposes a novel pipeline for extracting, relating and recommending scientific items from CORD-19 dataset, using entities from various ontologies: Gene Ontology (GO)[3], Disease Ontology (DO)[4], Human Phenotype Ontology (HP)[5], and Chemical Entities of Biological Interest Ontology (ChEBI)[6]. The use of ontologies for the NER phase allows the extraction of the entities from the text and the linking of the entities to a definition, avoiding the ambiguity of the terms. We selected these ontologies for their importance in the COVID-19 disease. With these, we may find drugs, genes, phenotypes and diseases related to COVID-19, which may guide the researchers in the discovery of new information for stopping the disease.

The main contributions of this work are:

- A dataset of 9k articles automatically annotated with relevant items/concepts for CORD-19;

- A sample dataset curated for CORD-19;

- A sample dataset with relations between the entities of the four ontologies;

- An implicit feedback matrix based on the previous datasets.

The source code for this work is fully available at: https://github.com/lasigeBioTM/knowledge-extraction-from-CORD-19.

## 2 Related work

Despite the novelty of CORD-19 dataset, the number of works being published using this dataset increases by the day. CORD-19 is being used for developing tools in various fields. (Tang et al., 2020) created a dataset for Question and Answering about COVID-19. (Kroll et al., 2020) developed a pipeline for creating a dataset based on biomedical NER, for chemicals, diseases, genes and species, using TaggerOne and GNormPlus tools. (Zhang et al., 2020) focused on a search engine for CORD-19 based on neural networks, the Neural Covidex. (Wang et al., 2020b) created the tool EVIDENCEMINER, which allows a user to introduce a sentence in natural language and to retrieve an evidence for that statement. They applied the EVIDENCEMINER to CORD-19. (Wang et al., 2020c) created the CORD-NER dataset, a dataset with entities from 75 fields, including genes, chemicals, diseases, and specific entities related to COVID-19, for example, coronaviruses, viral proteins, evolution, materials, substrates and immune responses. (Rahdari et al.) is developing a personalized exploratory search system for COVID-19, based on CORD-19, the CovEx. According to the authors, the system allows the user to search for keywords, recommending other keywords and research articles. The recommended keywords are extracted only from the title and abstract, using Bi-LSTM-CRF technique.

Not related to CORD-19, other research works created recommendation datasets for scientific fields. (Ortega et al., 2018) created a recommendation dataset of implicit feedback for the field of artificial intelligence with the format of <article,topic,cardinality>. Their work extracts topics related to artificial intelligence from articles and

---

[3] http://geneontology.org/
[4] https://disease-ontology.org/
[5] https://hpo.jax.org/app/
[6] https://www.ebi.ac.uk/chebi/

the cardinality is calculated according to the importance of the topic in the article. Then, the dataset is used for recommending topics and articles. The topics are extracted from the research articles using text mining techniques based on the articles' token frequency. They do not use NER techniques.

In (Barros et al., 2019), the authors developed a methodology called LIBRETTI to create recommendation datasets of implicit feedback for scientific fields, for recommending scientific entities, such as clusters of stars and Chemicals compounds. The methodology consists of given a list of scientific entities, finding articles related to these entities, and extracting the authors. The dataset has the format of <author,entity,rating>. The ratings are the number of articles a unique author wrote about an entity. This work uses the CHEBI ontology to extract the list of entities for a dataset of Chemical compounds; nevertheless, it is limited to this ontology and performs neither NER nor RE.

The objective of our pipeline is to create a tool for performing NER of multiple scientific fields in the CORD-19 dataset, followed by RE, and the creation of a recommendation dataset of implicit feedback based on LIBRETTI methodology, to recommend entities of different fields to the users/researchers. We hypothesize that using multiple ontologies in the creation of the recommendation dataset leads to an improvement in the performance of state-of-the-art CF algorithms, in particular when comparing with datasets with only one ontology.

## 3 Methodology and Experiments

The general workflow of the proposed pipeline is represented in Figure 1.

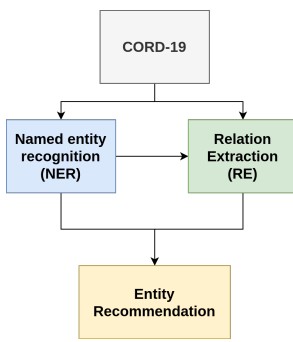

Figure 1: General pipeline.

The input of the pipeline is a dataset of research articles. First, we apply NER techniques for recognizing relevant entities in the text, using ontologies for linking the entities. Second, we extract the relations between the entities (RE). And third, we create an implicit feedback matrix and apply RS algorithms for recommending the recognized entities.

The dataset used in this work is the COVID-19 Open Research Dataset (CORD-19)(Wang et al., 2020a), more concretely, the 2020-03-13 version. The total number of documents is 29,500. For this work, we used the commercial use subset of 9,000 full-text documents.

### 3.1 Named Entity Recognition

To obtain ontology concepts from the CORD-19 dataset, we used the MER tool (Couto and Lamurias, 2018), which can identify entities in text from any ontology. We selected four biomedical ontologies to use as lexicons with MER so that we could identify those concepts in the texts: GO, HPO, DO and ChEBI. We used the latest version of each ontology available in June 2020. MER has the advantage of performing Named Entity Recognition and Entity Linking simultaneously, so we could obtain directly the reference ontology URI, which is necessary for the RS. Furthermore, it does not require annotated training data to identify new entity types.

The MER tool indexes the ontology labels and synonyms and uses regular expressions, therefore being limited in terms of what expressions it can identify. To assess the annotation quality, we manually evaluated a sample of 90 paragraphs from the CORD-19 dataset with four experts. The paragraphs were randomly selected from a pool of 100 documents, including paragraphs from the abstract, body and figure and table captions with at least five entities annotated by MER. We asked the expert annotators to verify the automatic annotations, modify and delete them, and add new annotations if necessary. Of the 90 paragraphs, 10 were annotated simultaneously by the four annotators to calculate the Inter-Annotator Agreement, using Fleiss' kappa (McHugh, 2012). Afterwards, we calculated Precision (Equation 1), Recall (Equation 2) and F1-score (Equation 3) metrics on the automatic annotations of these 90 paragraphs, given by the following formulas:

$$P = \frac{TP}{TP + FP} \qquad (1)$$

$$R = \frac{TP}{TP + FN} \qquad (2)$$

$$F1 = \frac{2PR}{P + R} \quad (3)$$

where TP corresponds to the total of True Positives (entities identified correctly), FP corresponds to False Positives (entities identified incorrectly) and FN corresponds to False negatives (entities that should have been identified).

We created a consensus corpus by merging the annotations of all 4 annotators. On the 10 overlapping paragraphs, we accepted each annotation if two or more annotators agreed on it. We calculated micro scores by adding the TP, FP and FNs of every paragraph and using Equations 1,2, and 3, and macro scores, which were the average of the Precision, Recall and F1-scores of all paragraphs.

## 3.2 Relation Extraction

We took initial steps towards COVID-19-related relation extraction training data (RE), providing a small sample dataset of ten documents, with all possible relationships between the four types of entities identified by our NER pipeline. Thus, we were able to establish ten different types of relations, encompassing the four ontologies (ChEBI, DO, HPO, and GO) in two machine-readable formats (XML and TSV), following previous works by Herrero-Zaro et al. (Herrero-Zazo et al., 2013) and Li et al. (Li et al., 2016), respectively.

## 3.3 Recommender System

For the creation of the RS dataset, we used a methodology called LIBRETTI (Barros et al., 2019). The methodology consists of creating datasets with the standard format of <user, item, rating>. The items are scientific entities and the users are authors from research articles, where these items are mentioned. The items may be obtained, for example, from a list or, as in previous work, from an ontology (Barros et al., 2019, 2020). As previously mentioned, for this work we used items/entities from four distinct ontologies: chemical compounds from CHEBI, functions of genes from GO, phenotypic abnormalities from HP, and diseases from DO. The output from this phase is a recommendation dataset of <user, item, rating>, where the users are authors from research articles, the items are entities from CHEBI, GO, HP or DO, and the ratings are the number of articles an author wrote about an entity. Our goal was to evaluate if using more ontologies, i.e., increasing the number

of entities for each author, the results of the recommendation algorithms are better for the same users. Thus, we assessed the results in the dataset with the items from all the ontologies, and with the items of each ontology alone, for the same group of users. We consider as baseline the results obtained with the datasets containing only items from each single ontology.

The recommendation datasets are evaluated using offline evaluation methods for the quality of the recommended ranked list of items (Shani and Gunawardana, 2011). From the vast range of metrics for evaluate ranked lists, we selected Precision@k (Equation 4), Recall@k (Equation 5) and Mean Reciprocal Rank (MRR)@k (Equation 6), where k is the size of the recommendation list.

$$Precision@k = \frac{relevant\_items@k}{k} \quad (4)$$

$$Recall@k = \frac{relevant\_items@k}{total\_relevant\_items} \quad (5)$$

$$MRR = \frac{1}{n\_users} \sum_{i=1}^{n\_users} \frac{1}{rank\_i} \quad (6)$$

Precision@k is a measure of the relevant items recommended in the top@k list, recall@k the number of relevant items recommended in the top@k list, and MRR evaluates in which position the first relevant item appears. All evaluation metrics range between 0 and 1, with the best values being closest to 1.

Since the dataset consists of ratings obtained by implicit feedback, we selected Alternating Least Squared (ALS)[7] (Hu et al., 2008), a recommendation algorithm capable of dealing with implicit feedback datasets. ALS is a latent factor algorithm that addresses the confidence of a user-item pair rating. The ALS goal is to minimize the least-squares of the rating matrix and the matrix resultant from the dot product of the user matrix and item matrix. ALS is also suitable for recommending ranked lists of items. This algorithm was already used in similar datasets with positive results, for recommending Chemical Compounds (Barros et al., 2020). The datasets for the evaluation were split in 80% of users and items for training and 20% for testing.

---

[7] https://implicit.readthedocs.io/en/latest/index.html

## 4 Results and Discussion

### 4.1 Named Entity Recognition

The entity annotation part of the pipeline obtained a total of 2,412,671 entity mentions on the comm_use_subset of the CORD-19 dataset (9k documents). Table 1 shows the counts of the entities obtained according to each ontology.

Table 1: Statistics of the entities obtained on the CORD-19 commercial subset of 9k documents.

| Ontology | Total mentions | Unique mentions |
| --- | --- | --- |
| CHEBI | 1,302,219 | 6,693 |
| GO | 484,266 | 3,258 |
| DO | 314,959, | 1,726 |
| HP | 311,227 | 1,774 |
| Total | 2,412,671 | 13,451 |

We obtained an average of 268.07 entity mentions per document and 67.02 unique concepts per document. The results of our manual evaluation are provided in Table 2. Our gold standard of 90 paragraphs obtained an IAA of 0.2978, which indicates fair agreement, according to (Landis and Koch, 1977). However, if we do not take into consideration the ontology URIs, this agreement rises to 0.3760. This indicates that the definition of the URIs was a source of ambiguity and the annotators did not always agree on what was the best ontology concept for a named entity. The Precision, Recall and F1-score values obtained indicate that the entities were mostly correctly identified, with a relatively high macro and micro Recall value. It is also possible to observe that the highest F1-scores obtained were with the gold standard where at least two annotators had to agree to accept an annotation.

The positive results obtained in this phase of the pipeline allows us to use the NER dataset for subsequent Relation Extraction, and for the creation of the recommendation dataset.

### 4.2 Relation Extraction

To accomplish the RE dataset, we only considered relations between entities in the same text portion, following the original dataset, identified common NER errors to exclude those entities from participating in relations, and did not consider relations between the same entities in different places of the text portion. The resulting final counts are presented in Table 3.

Following, we present Examples 4.1 and 4.2 of relations extracted from text.

**Example 4.1** *Several viruses use classical receptors and transmembrane proteins that are widely represented in cells and are not restricted to the monocyte/macrophage population, such as nucleolin by the respiratory syncytial virus [75] ; sialic acid sugars by the influenza virus [76], mouse hepatitis virus [77] and Theiler's murine encephalomyelitis virus [78; and phosphatidylserine by the vesicular stomatitis virus [79].*

*sialic acids (CHEBI_26667) - influenza (DOID_8469)*

*sialic acids (CHEBI_26667) - hepatitis (HP_0012115)*

*sialic acids (CHEBI_26667) - hepatitis (DOID_2237)*

*sialic acids (CHEBI_26667) - encephalomyelitis (DOID_640)*

*phosphatidylserine (CHEBI_18303) - stomatitis (DOID_9637)*

*phosphatidylserine (CHEBI_18303) - stomatitis (HP_0010280)*

**Example 4.2** *For receptor-mediated entry, viruses can employ both nonspecific receptors, where a virus accesses a broad range of cell populations, or highly specific interactions between the virus and cell surface receptors, where a virus infects a limited set of target cells; this determines the tropism of viral infection.*

*cell surface (GO_0009986) - tropism (GO_0009606)*

*tropism (GO_0009606) - viral infection (DOID_934)*

*tropism (GO_0009606) - viral infection (GO_0016032)*

In both Examples 4.1 and 4.2, we present sentences from a research article and the respective relations between the entities extracted from these sentences. For example, in Example 4.1 we identified relations between the chemical compound *sialic acids* (CHEBI_26667) and the disease *influenza* (DOID_8469).

We believe this to be an initial step towards COVID-19-related relation extraction training data. The following step should be running existing machine learning models (Sousa and Couto, 2020) to classify the possible relations as true or false. Finally, we intend to use the RE information as data for the RS detailed below.

Table 2: Results of the manual evaluation of the NER module. Min Votes corresponds to the number of annotators necessary to agree on the gold standard annotations.

| Min Votes | Micro | | | Macro | | |
| --- | --- | --- | --- | --- | --- | --- |
| | P | R | F1 | P | R | F1 |
| 1 | 0.7740 | 0.8007 | 0.7871 | 0.7671 | 0.827 | 0.7601 |
| 2 | 0.7656 | 0.8211 | 0.7924 | 0.761 | 0.8411 | 0.7641 |
| 3 | 0.7586 | 0.8255 | 0.7907 | 0.7532 | 0.8423 | 0.7616 |
| 4 | 0.7457 | 0.8374 | 0.7889 | 0.7408 | 0.8542 | 0.7579 |

Table 3: Statistics for the relation extraction sample dataset possible relations.

| Pair | Count |
| --- | --- |
| GO-GO | 410 |
| GO-CHEBI | 765 |
| GO-HP | 396 |
| GO-DO | 342 |
| CHEBI-CHEBI | 489 |
| CHEBI-HP | 457 |
| CHEBI-DO | 349 |
| HP-HP | 242 |
| HP-DO | 440 |
| DO-DO | 149 |
| Total | 4,039 |

## 4.3 Recommender System

From the methodology presented in Section 3.3, we obtained a recommendation dataset for recommending the biomedical entities in CORD-19, previously extracted with NER (Section 3.1), called CORD-19 Recommendation Dataset (CORD-19-RD). CORD-19-RD was assessed for the items in all four ontologies (CORD-19-RD-all), and sampled by ontology, in order to evaluate how the use of items from the four ontologies influences the results when compared to the individual ontologies. Thus, we have the sampled datasets CORD-19-RD-chebi, CORD-19-RD-go, CORD-19-RD-hp and CORD-19-RD-do. Table 4 shows the statistics of CORD-19-RD and its samples, presenting the number of users, items and ratings, the sparsity of each dataset, the maximum and minimum rating values ( max and min), and also the mean of items by user (Mean items), and the mean of users by item (Mean users).

The number of users in the various datasets remains almost the same, with the highest variation in the CORD-19-RD-go. The number of items decreases drastically from CORD-19-RD-all to CORD-19-RD-do, with a variation of 11.644 items.

The number of ratings also decreases from CORD-19-RD-all to CORD-19-RD-do. Despite the decrease in the number of items, the sparsity is not affected. Notwithstanding that, the mean of items by user is much higher for CORD-19-RD-all.

Figure 2 shows the results of applying the ALS algorithm to the different datasets CORD-19-RD-all, CORD-19-RD-chebi, CORD-19-RD-go, CORD-19-RD-hp and CORD-19-RD-do, for Precision@k, Recall@k, and MRR@k, with k varying from 1 to 20, with steps of 1.

Analyzing Figure 2, ALS best performs in CORD-19-RD-all dataset for all the evaluation metrics. Looking at these results and the values in Table 4, we can see the relation between the mean of items by user and the results of ALS. The higher the mean of items by user, the higher the results for all the evaluation metrics for the same number of users. The presented results prove our hypothesis that items from several ontologies, i.e., from more than one field of Science, improve the results of state-of-the-art recommendation algorithms. Our pipeline solves the lack of ratings and allows the recommendation of items from various fields, allowing the development of multidisciplinary RS. For the COVID-19 case study, we will be able, for example, to recommend chemical compounds to some user interested in the disease, which may increase the study of new drugs, that otherwise would be much harder to find the connection.

Table 5 shows an example of the top@20 recommendation for a user in the CORD-19-RD-all. The recommendation algorithm recommends items from all the ontologies for this user, even though she/he does not have all the ontologies represented in the training set. This may lead to the discovery of new diseases similar to COVID-19, and chemicals used in the treatment of those diseases, which may be an object of study for its use in COVID-19.

The next step is to use the relation extracted in the RE phase to recommend the relations between

Table 4: Statistics for the dataset CORD-19-RD-all, CORD-19-RD-chebi, CORD-19-RD-go, CORD-19-RD-hp and CORD-19-RD-do.

| Dataset | Users | Items | Ratings | Sparsity | max | min | Mean items | Mean users |
|---|---|---|---|---|---|---|---|---|
| CORD-19-RD-all | 45,401 | 13,353 | 3,888,870 | 0.993 | 39 | 1 | 85.6 | 291.2 |
| CORD-19-RD-chebi | 45,401 | 6,642 | 1,939,568 | 0.993 | 39 | 1 | 42.7 | 292.0 |
| CORD-19-RD-go | 44,646 | 3,250 | 800,135 | 0.994 | 31 | 1 | 17.9 | 246.1 |
| CORD-19-RD-hp | 45,343 | 1,752 | 684,332 | 0.991 | 37 | 1 | 15.1 | 390.6 |
| CORD-19-RD-do | 45,041 | 1,709 | 464,835 | 0.993 | 39 | 1 | 10.3 | 271.9 |

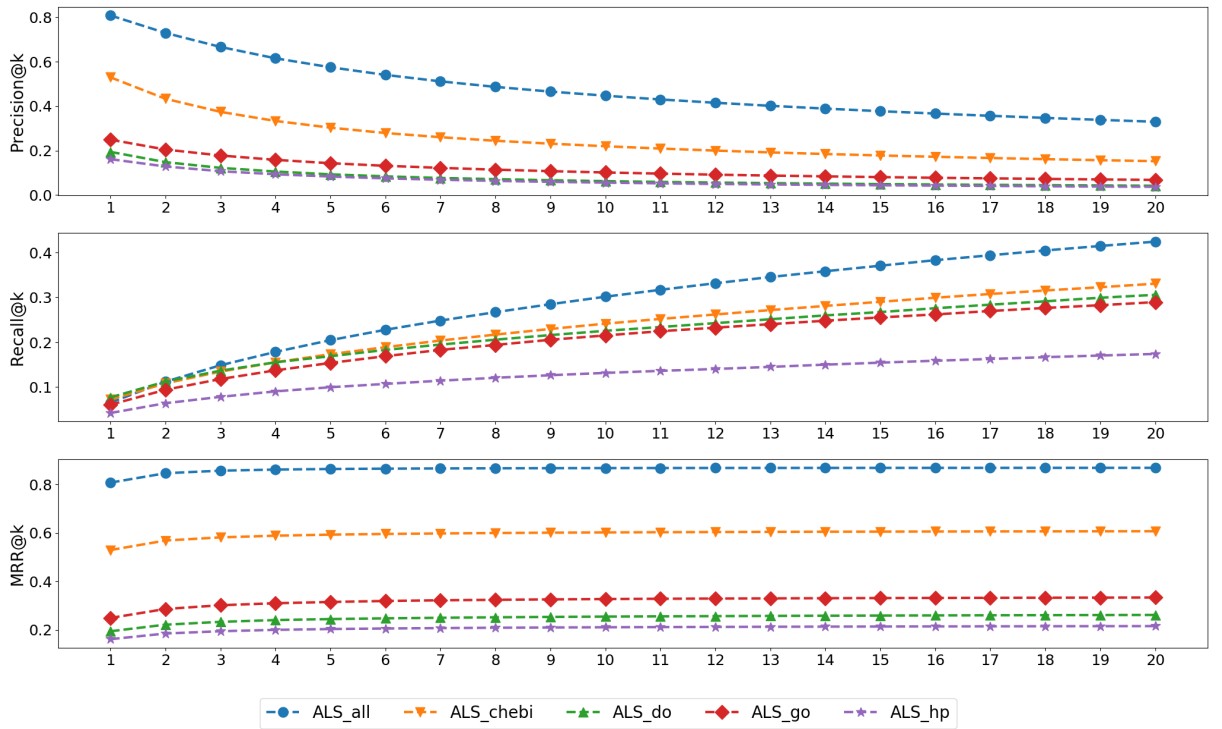

Figure 2: Results of the algorithm ALS for Precision@k, Recall@k and MRR@k, applied to CORD-19-RD-all, CORD-19-RD-chebi, CORD-19-RD-go, CORD-19-RD-hp and CORD-19-RD-do.

the items and create an explainable RS. For example, we could use the relations extracted in Examples 4.1 and 4.2 for creating knowledge graphs and recommending items from different fields related to the articles.

We still need to understand if all the entities are suitable for being recommended. For example, the entity DOID_4 (Disease) is one of the most identified in the NER phase, consequently being one of the most recommended in the recommendation phase. However, is it relevant for a user the recommendation of "disease"? We are now studying new methods for assigning relevance to each entity. Additionally, we can extend this approach to other documents from the CORD-19, as new versions are released.

The code for this work is fully available at: https://github.com/lasigeBioTM/

knowledge-extraction-from-CORD-19.

## 5 Conclusion

Given the growing number of publications, this work's goal was to develop a pipeline for extracting biomedical entities from scientific literature, finding the relations between them, and recommending entities of interest for a particular researcher. The second goal was to prove that by using ontologies from different science fields, CF RS would achieve better results when recommending ranked lists of entities to the users. We used as a case study the CORD-19 dataset, which is a dataset in a field of high relevance for this Era. Using this dataset, we performed NER using four ontologies, CHEBI, DO, HP, and GO, creating an annotated dataset of 9k documents. We also curated 100 documents

Table 5: Example of recommendation for a user in the CORD-19-RD-all. The green cells are the relevant items recommended.

| Ontology ID | Name |
|---|---|
| CHEBI_17076 | streptomycin |
| GO_0019012 | virion |
| CHEBI_149681 | methcathinone |
| HP_0001903 | Anemia |
| CHEBI_25212 | metabolite |
| CHEBI_17234 | glucose |
| GO_0019079 | viral genome replication |
| CHEBI_55308 | poly(2,5-furan) macro-molecule |
| DOID_0050639 | primary cutaneous amyloidosis |
| CHEBI_15366 | acetic acid |
| CHEBI_33601 | safranin O |
| CHEBI_53233 | 3-(4,5-dimethylthiazol-2-yl)-2,5-diphenyltetrazolium bromide |
| CHEBI_15756 | hexadecanoic acid |
| DOID_3247 | rhabdomyosarcoma |
| GO_0006631 | fatty acid metabolic process |
| HP_0040280 | Obligate |
| GO_0005886 | plasma membrane |
| CHEBI_17544 | hydrogencarbonate |
| CHEBI_32952 | amine |
| HP_0030078 | light-harvesting complex, core complex |

from this dataset, achieving positives results for precision, recall, and F1-score. The RE phase is in the beginning. Nevertheless, this is a first step for creating a full dataset of relations between the fields in study, which can then be used for generating a knowledge base for COVID-19. We created a dataset with more than 3 Million ratings, 45 thousand users, and 13 thousand items from four relevant scientific fields in the recommendation phase. We concluded that using items from several fields for the same users, the CF algorithm reached better results. For future work, we intend to increase the number of research documents, the number of documents manually annotated, and provide a better baseline. Furthermore, the next step is to integrate the RE dataset in the RS. It is also important to perform online tests for a better understanding of the relevance of recommended items. For such, an online recommendation platform will be developed.

## Acknowledgments

This work was supported by the Fundação para a Ciência e Tecnologia (FCT), under LASIGE Strategic Project UIDB/00408/2020, CENTRA Strategic Project UIDB/00099/2020, FCT funded project PTDC/CCI-BIO/28685/2017, PhD Scholarship SFRH/BD/128840/2017 and PhD Scholarship, ref. SFRH/BD/145221/2019.

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
