# OpenReview forum: "COVID-19: A Semantic-Based Pipeline for Recommending Biomedical Entities"
_EMNLP/2020/Workshop/NLP-COVID — NLP-COVID19-EMNLP Poster_

### Official Review · AnonReviewer2 · 2020-09-22
**Deployment of recommender system based on multiple NER ontologies**

**Rating:** 6
**Confidence:** 1

**Review:**

The authors develop datasets for NER and RE extraction of COVID-19 literature. They used the generated data to generate concept recommendations.

Reasons to accept:
- The developed datasets might be valuable to further researchers

Reasons to reject:
- I do not understand why would it be useful to recommend concepts to someone. It might be because of my limited expertise on the field.

---

> ### Author Response · Authors · 2020-09-28
> **RE: Deployment of recommender system based on multiple NER ontologies**
>
> Dear Reviewer2,
>
> We are very grateful for your review.
>
> About your concern, we commit to make it more clear in the camera-ready version that we do not want to recommend concepts, we want to recommend specific biomedical entities, such as chemical compounds, which may be useful for the researchers for their studies and that otherwise, with the large amount of papers and entities, would be much harder to discover.

---

### Official Review · AnonReviewer1 · 2020-09-24
**An interesting initial approach to recommendation systems in academic domains. The evaluation requires additional work.**

**Rating:** 6
**Confidence:** 2

**Review:**

**Core review**:

This paper proposes a pipeline for the automatic extraction of entities and relations from the CORD-19 dataset and recommendation of relevant entities in those papers. The authors apply a system (MER, Couto and Lamurias, 2018) for the automatic extraction of entities which is seeded with 4 ontologies in the biomedical domain. A total of 9000 documents from CORD-19 are automatically annotated with entities, from which 100 documents are manually curated and a small subset is used to evaluate the inter-annotator agreement and manually annotate relevant relations (relations are defined in the ontologies used). Afterwards, a recommendation technique (LIBRETTI) is applied by considering author-entity-mention counts as the user-item-rating triplets. Thus, as I understand, the overall pipeline proposed has the purpose of suggesting an author of a paper in the annotated dataset with novel entities that might be of interest for that author and not mentioned in their papers.

The main hypothesis of the paper is that the use of ontologies from different domains for the NER phase improves the quality of the recommendation. While this hypothesis is effectively demonstrated by the experimental results, I believe this is a direct consequence of the fact that these ontologies are from different domains and thus mostly disjoint, which in my opinion renders the hypothesis self-evident. With respect to the results, the significance of the metrics provided is hard to evaluate without some comparison with alternative systems or at least some baselines.

Although I consider there is value in the proposal, since building such a recommendation system at a large scale (e.g., at the scale of Semantic Scholar or Google Scholar) would be extremely valuable for researchers, I still consider a significant effort is necessary both to improve the extraction and recommendation procedures, but also regarding manual annotation, evaluation, and the actual development of such a tool. Finally, I commend the authors for their effort. I recommend working on improving the motivation of the research; at least personally, it was hard for me to understand the underlying motivation for recommending entities, and I think this is a very important point.

**Reasons to accept:** The automatically and manually annotated corpora can be valuable resources for the community. Also, the source code is provided, which improves reproducibility and fosters collaboration.

**Reasons to reject:**  The significance of the results is hard to evaluate since there is no comparison with alternatives or baselines.

**Other considerations:**

- About the phrase "an IAA of 0.2978 which indicates fair agreement" -- It is my understanding that a fair Kappa agreement is considered above 0.70 unless the authors are using a different formulation, in which case I would recommend the authors to explicitly state and justify the threshold above which they consider the agreement "fair".

- The Precision, Recall and F1 scores are considered "high" by the authors, however, since there is no baseline or comparison it is hard to determine the significance of these measures.

- Recurrent typos in page 2: *"¡article,topic,cardinality¿"*, *"¡author,entity,rating¿"*, in Page 4: *"¡user, item, rating¿"*, ... (I believe this stems from the incorrect to use of $<$ and $>$ in LaTeX)

---

> ### Author Response · Authors · 2020-09-28
> **RE: An interesting initial approach to recommendation systems in academic domains. The evaluation requires additional work.**
>
> Dear Reviewer1,
>
> Thank you for reading our paper and for the careful analysis of it.
>
> Regarding your concerns:
> - We commit to better explain the difficulty to compare without baseline, since we wanted to test if the use of more ontologies in the NER phase would improve the results of the recommender system, we assumed as baselines each dataset with the single ontologies.
> - We also commit to add a clear motivation, by stating that there are hundreds of research documents being published every day, with important information enclosed inside. It is easy to get lost with all this content. Thus, what our system aims is to provide a toll which extracts important content from the documents using NER, and recommends this content to the users. We use the ontologies because like this we can link the entities in the documents to platforms with more information about each entity.
> - If accepted, we also agree and commit to better describe the relevance of recommending entities. There are several systems recommending research articles, but few recommending scientific items. For example, you are a researcher interested in a chemical compound and you are searching for information about this. What our recommender system does is, based on your interests and based on the interests of your peers, recommend other chemical compounds which may be of interest as well for your research.

---

### Official Review · AnonReviewer3 · 2020-09-25
**Need to improve the experiment design**

**Rating:** 4
**Confidence:** 4

**Review:**

This paper proposes a system for biomedical entity recommendation. The system performs entity recommendation in a pipeline manner with three steps: named entity recognition, relation extraction, and entity recognition. The paper shows that the recommendation accuracy can be enhanced if multiple ontologies are used for recognising and linking the entities. The paper is well-written and easy to follow

My major concern is that whether the experimental results provide sufficient evidence to support the hypothesis that the authors formulate: if multiple different ontologies are used for recognising entities, the accuracy of entity recognition increases. As shown in Figure 2, the accuracy of using all ontologies (ALS-all) outperforms any other dataset with only one ontology (e.g., ALS-chebi which only includes entities in CHEBI ontology). This might be a positive sign showing that other ontologies are contributing to model’s understanding about how to recommend CHEBI entities. However, it might be also simply because of the model’s learning curve: the data in ALS-all is a super set of ALS-CHEBI, and the model simply learns more with more available data, no matter which ontology the extra data comes from. And this can be evident by that the accuracy in Figure 2 (ALS-all>ALS-chebi>ALS-go>ALS-hp~=ALS-do) is actually roughly proportional to the data set size if we look at the size of each data set in Table 4. As such, the authors should consider to another baseline scenario where the entities are randomly sampled, without considering the ontologies.

Secondly, the rating matrix is not implicit feedback. Implicit feedback means the scenario where rating is not available and only users’ actions are available. In implicit feedback setting, the interaction between users and items is only recorded as 0 or 1 (interacted or not). Therefore, pairwise ranking is pervasively used for learning. The ALS algorithm is usually preferred in the setting of explicit feedback, rather than implicit feedback.

Reasons to accept: well-written, data set can be useful
Reasons to reject: the design of the experiments cannot fully support the claim in the paper

---

> ### Author Response · Authors · 2020-09-28
> **RE: Need to improve the experiment design**
>
> Dear Reviewer3,
> Thank you for your careful and detailed review.
>
> We agree with your first concern, that the experimental results may not be sufficient for fully supporting the proposed hypothesis, so if the article is accepted we commit to make the limitations of our proposal more clear, and how we are planning to improve by  working on expanding the dataset and the number of papers, the number of documents manually annotated and also a better baseline. Nevertheless, we believe that our pipeline may help the researchers in the discovery of useful biomedical concepts hidden inside the research documents.
>
> About your second concern, we believe that the notion of implicit feedback was not clearly perceived by the reviewer, for example the duration that a user watches a video is implicit feedback it is not only recorded as 0 or 1 (interacted or not). However, we accept that the definition was not clear and because of that we commit to define it in a better way, i.e. assuming that implicit feedback is all the feedback that is not explicitly provided by the users of a platform, such as in a five star system, for example.
> In regard to ALS, according to previous work, it is an algorithm which achieved good results in datasets similar to the ones presented in this study, so we commit to add references that explicitly address this issue in the camera-ready version of the paper, if accepted.